# Geo-Neus: Geometry-Consistent Neural Implicit Surfaces Learning for Multi-view Reconstruction

**Qiancheng Fu**[1*]     **Qingshan Xu**[2*]     **Yew-Soon Ong**[2,3]     **Wenbing Tao**[1†]

[1]Huazhong University of Science and Technology     [2]Nanyang Technological University
[3]A*STAR, Singapore
[1]{fqc98,wenbingtao}@hust.edu.cn [2]{qingshan.xu,asysong}@ntu.edu.sg

## Abstract

Recently, neural implicit surfaces learning by volume rendering has become popular for multi-view reconstruction. However, one key challenge remains: existing approaches lack explicit multi-view geometry constraints, hence usually fail to generate geometry-consistent surface reconstruction. To address this challenge, we propose geometry-consistent neural implicit surfaces learning for multi-view reconstruction. We theoretically analyze that there exists a gap between the volume rendering integral and point-based signed distance function (SDF) modeling. To bridge this gap, we directly locate the zero-level set of SDF networks and explicitly perform multi-view geometry optimization by leveraging the sparse geometry from structure from motion (SFM) and photometric consistency in multi-view stereo. This makes our SDF optimization unbiased and allows the multi-view geometry constraints to focus on the true surface optimization. Extensive experiments show that our proposed method achieves high-quality surface reconstruction in both complex thin structures and large smooth regions, thus outperforming the state-of-the-arts by a large margin. Code: `https://github.com/GhiXu/Geo-Neus`.

## 1   Introduction

Reconstructing surfaces from calibrated multi-view images is a long-standing problem in computer vision and graphics. In the past years, traditional methods [30, 37, 15, 17] have adopted a multi-step pipeline to achieve impressive reconstruction results. Such a pipeline requires depth maps or point clouds to generate surface meshes. These intermediate representations inevitably introduce accumulated errors for the final reconstructed geometry. Recently, directly reconstructing surfaces from images [25, 43, 35, 42, 26] has attracted great interest for its potential to alleviate the accumulated errors and produce high-quality reconstructions. To achieve this, existing approaches represent surfaces as neural implicit representations and leverage volume rendering [21] to optimize them.

Inspired by neural volume rendering [23, 45] that simultaneously learns volume density and radiance field from input images, recent works [35, 42] use signed distance functions (SDF) [27] for surface representation and introduce the SDF-induced density function to enable the volume rendering to learn an implicit SDF representation. In essence, these works still focus on direct color field modeling by volume rendering integral rather than explicit multi-view geometry optimization. Therefore, existing approaches usually fail to generate geometry-consistent surface reconstruction. Intuitively, volume rendering samples multiple points along each ray and expresses the output pixel colors as the integral of the radiance field, or the weighted sum of sampled colors along the ray (cf. Fig. 1(a)). It means that the volume rendering integral directly optimizes the integral of geometry instead of the

---

[*]Equal contribution.

[†]Corresponding author.

36th Conference on Neural Information Processing Systems (NeurIPS 2022).

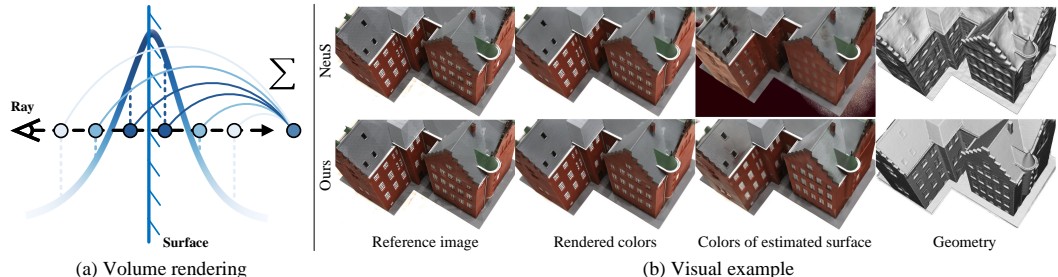

| (a) Volume rendering | (b) Visual example |
| --- | --- |

Figure 1: (a) Illustration of volume rendering. (b) A visual example. The volume rendering in NeuS uses integral colors to implicitly supervise surface modeling. Although its rendered colors achieve good results, the colors of estimated surface fail to preserve object geometry information. This shows the bias between rendered colors and geometry. In contrast, our approach achieves structure-preserving colors of estimated surface and produces geometry-consistent surface reconstruction.

single surface intersection along the ray. This obviously introduces bias for geometry modeling, thus hindering true surface optimization. In Fig.1(b), we show the reconstruction case of NeuS [35], in which the bias between rendered colors and object geometry can be observed intuitively. Rendered colors are obtained by the color network via volume rendering. Surface colors are formed by the predicted colors of the surface where the SDF values are zeroes. It can be easily seen that there exists a gap between the rendered colors and the surface colors. Thus the reconstructed surface is imprecise despite the high-quality rendered image, indicating the bias between the color rendering and implicit geometry. (Detailed theoretical analysis will be elaborated later).

To address the above problem, we propose Geo-Neus to devise an explicit and accurate neural geometry optimization model for geometry-consistent neural implicit surfaces learning by volume rendering, leading to better multi-view 3D reconstruction. Specifically, we directly locate the zero-level set of SDF networks and explicitly perform multi-view geometry optimization by leveraging the sparse geometry from structure from motion (SFM) and photometric consistency in multi-view stereo. This model has several benefits. First, directly locating the zero-level set of SDF networks guarantees that our geometry modeling is unbiased. This enables our method to focus on true surface optimization. Second, we show that explicitly enforcing multi-view geometry constraints on the located zero-level set of SDF networks allows our method to generate geometry-consistent surface reconstruction. Previous neural implicit surfaces learning mainly uses the rendering loss to implicitly optimize SDF networks. This results in geometry ambiguity during the training optimization. Our introduced two types of explicit multi-view constraints encourage our SDF networks to reason about the correct geometry, including both complex thin structures and large smooth regions.

In summary, our contributions are: **1)** We theoretically analyze that there exists a gap between volume rendering integral and point-based SDF modeling. This demonstrates that it is necessary to directly supervise the SDF networks to boost the neural implicit surfaces learning. **2)** Based on our theoretical analysis, we propose to directly locate the zero-level set of SDF networks and leverage multi-view geometry constraints to explicitly supervise the training of SDF networks. In this way, the SDF networks are encouraged to focus on true surface optimization. Extensive experiments further validate the effectiveness of our theoretical analysis and the proposed direct optimization of SDF networks. We show that our proposed Geo-Neus is capable to reconstruct both complex thin structures and large smooth regions. Therefore, it greatly outperforms the state-of-the-art surface reconstruction methods, including traditional methods and neural implicit surface learning methods.

## 2 Related work

**Traditional multi-view 3D reconstruction.** Traditional multi-view 3D reconstruction is the classical pipeline of surface reconstruction from multi-view images. Given multi-view input images, traditional multi-view 3D reconstruction uses structure from motion (SFM) [33, 29] to extract and match features of neighbor views, and estimate camera parameters and sparse 3D points. After that, multi-view stereo (MVS) [30, 9, 37, 38, 36] is applied to estimate dense depth maps for each view and then all the depth maps are fused into dense point clouds. Finally, the surface reconstruction method [15, 17, 6], e.g., screened Poisson Surface Reconstruction [15] is used to reconstruct surfaces from point clouds. Traditional methods have achieved great success on various occasions, but there

exists incompleteness of surface in some cases because their multiple intermediate steps are not made into an ensemble. With the development of deep learning, many attempts have been made on learning-based multi-view reconstruction [14, 40, 39, 27, 22], but the problem still exists.

**Implicit representation of surface.** Surface reconstruction methods can be generally divided into explicit methods and implicit methods, depending on the representation of surface. Explicit representation includes voxels [5, 31] and triangular mesh [3, 4, 16], which are limited by the resolution. Implicit representation uses an implicit function to represent the surface and thus is continuous. The surface can be extracted using the implicit function at any resolution. Traditional reconstruction methods, e.g., screened Poisson Surface Reconstruction [15], use basic functions to form the implicit function. As for learning-based methods, the most commonly used forms are the occupancy function [22, 28] and the signed distance function (SDF) [27] represented by the network. Based on these functions, many implicit surface reconstruction methods from point clouds have been proposed, e.g., ONet [22], DeepSDF [27], Point2Surf [8] and etc. For these methods, point clouds are obtained by scanner devices or multi-view stereo methods. That is, these point clouds are uniform and complete. Therefore, these methods are rarely applied on the sparse point clouds produced by SFM to reconstruct surfaces (In fact, we show that these methods degrade on sparse point clouds from SFM in our supplementary material). In this work, we use the sparse points from SFM as an explicit geometry supervision and show that they can facilitate the neural implicit surface reconstruction.

**Neural implicit surface reconstruction.** Neural implicit field is a new way to represent the geometry of objects. With NeRF [23] first using the neural radiance field represented by Multi-Layer Perceptron (MLP) in novel view synthesis, plenty of works [32, 18, 20] have sprung up using neural networks to represent scenes. IDR [43] reconstructs surfaces with neural networks by representing the geometry as the zero level set of an MLP that is considered to be an SDF. MVSDF [44] imports information from the MVS network to arrive at more geometry priors. VolSDF [42] and NeuS [35] use the weight function that involved SDF during the rendering process to make colors and geometry closer. UNISURF [26] explores the balance between surface rendering and volume rendering. The surface reconstructed by the neural network shows better completeness compared with the traditional multi-view reconstruction methods, especially when dealing with non-Lambertian cases. However, complex structures are not handled well. Meanwhile, flat planes and sharp corners could not be guaranteed. NeuralWarp [7], a concurrent work, also explores the use of patch-match on neural surface reconstruction. It combines volumetric rendering with a patch warping integration technique, which aggregates colors from points sampled along the camera ray from source views with patch warping. This way of patch aggregation is similar with volume rendering and shares the same sampled points and the same weights with those used by color integration. Note that NeuralWarp uses patch match with the color aggregation to optimize weights of sampled points, and thus to optimize the geometry indirectly. As we will analyze later, this kind of color integration operation will cause bias in the colors and the geometry. Therefore, NeuralWarp could not be trained from scratch and relies on the pre-trained model of VolSDF. Differently from NeuralWarp, our method locates the predicted surface of the SDF network using SDF-based interpolation and uses patch match to measure the photometric consistency among neighboring views. In this way, Geo-Neus can be trained from scratch and gets much better performance.

# 3 Method

Given posed multi-view images of an object, we aim at reconstructing the surface by neural volume rendering without mask supervision. The spatial field of the object is represented by a signed distance function (SDF), and the corresponding surface is extracted using the zero level set of the SDF. In the process of volume rendering, our goal is to optimize the signed distance function. In this section, we first analyze the inherent bias in color rendering which causes the inconsistency between rendered colors and implicit geometry. Then we introduce explicit SDF optimization to achieve geometry consistency. An overview of our approach is shown in Fig. 2.

## 3.1 Bias in color rendering

In the process of volume rending, there is a gap between the rendered colors and the geometry of the object. The rendered colors are not consistent with the real colors of the surface.

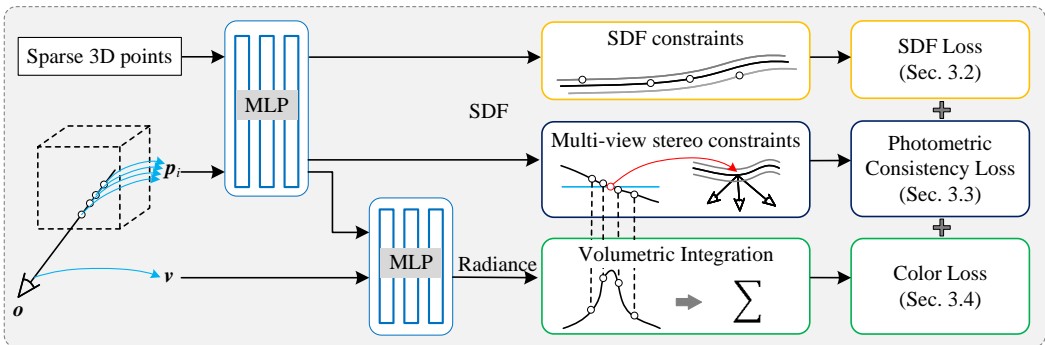

Figure 2: Overview of Geo-Neus. Previous neural implicit surfaces learning methods mainly depend on the color loss to implicitly supervise the SDF network. Our proposed Geo-Neus explicitly supervises the SDF network by introducing the SDF loss from sparse 3D points and photometric consistency loss from multi-view stereo.

For an opaque solid object $\Omega \in \mathbb{R}^3$, the opacity can be represented by an indicator function $\mathcal{O}(\boldsymbol{p})$:

$$\mathcal{O}(\boldsymbol{p}) = \left\{ \begin{array}{ll} 1, & \boldsymbol{p} \in \Omega \\ 0, & \boldsymbol{p} \notin \Omega \end{array} \right. . \tag{1}$$

When we see some colors or we capture some colors with cameras, the colors are the light that transfers along the light ray into our eyes or cameras. Based on the inherent optical properties of the opaque solid object, we approximately assume that the colors $C$ of image set $\{I_i\}$ are the colors $c$ of object intersecting with the light ray $\boldsymbol{v}$ from the corresponding camera position $\boldsymbol{o}$:

$$C\left(\boldsymbol{o}, \boldsymbol{v}\right) = c\left(\boldsymbol{o} + t^*\boldsymbol{v}\right), \tag{2}$$

where $t^* = \mathrm{argmin}\,\{t|\boldsymbol{o} + t\boldsymbol{v} = \boldsymbol{p},\ \boldsymbol{p} \in \partial\Omega,\ t \in (0,\ \infty)\}$. $\partial\Omega$ represents geometry surfaces. The assumption is appropriate because light that transmits through the opaque object can be omitted. The intensity of light decays to about zero drastically when passing through the surface of the opaque object. Let us represent the surface of the object mathematically with the signed distance function. The signed distance function $sdf(\boldsymbol{p})$ is the signed distance between a spatial point $\boldsymbol{p}$ and the surface $\partial\Omega$. In this way, the surface $\partial\Omega$ can be represented as:

$$\partial\Omega = \{\boldsymbol{p}|sdf\left(\boldsymbol{p}\right) = 0\}. \tag{3}$$

With neural volume rendering, we estimate the signed distance function $\hat{sdf}$ and color field $\hat{c}$ by Multi-Layer Perceptron (MLP) networks $F_\Theta$ and $G_\Phi$:

$$\hat{sdf}\left(\boldsymbol{p}\right) = F_\Theta\left(\boldsymbol{p}\right), \tag{4}$$

$$\hat{c}\left(\boldsymbol{o}, \boldsymbol{v}, t\right) = G_\Phi\left(\boldsymbol{o}, \boldsymbol{v}, t\right). \tag{5}$$

Thus the estimated colors of the image with camera position $\boldsymbol{o}$ can be represented as:

$$\hat{C} = \int_0^{+\infty} w\left(t\right) \hat{c}\left(t\right) dt, \tag{6}$$

where $t$ is the depth along the ray that comes from $\boldsymbol{o}$ with the direction $\boldsymbol{v}$ and $w(t)$ is a weight for the point at $t$. For simplicity, the notes $\boldsymbol{o}$ and $\boldsymbol{v}$ are omitted. To obtain discrete counterparts of $w$ and $\hat{c}$, we also sample $t_i$ discretely along the ray and use the Riemann sum:

$$\hat{C} = \sum_{i=1}^{n} w\left(t_i\right) \hat{c}\left(t_i\right). \tag{7}$$

Following NeuS [35], $w(t_i)$ is computed as $w(t_i) = T(t_i)\alpha(t_i)$, where $T(t_i) = \prod_{j=1}^{i-1}(1 - \alpha(t_j))$ represents accumulated transmittance, $\alpha(t_i) = \max(\frac{\Phi_s(sdf(\boldsymbol{p}_i)) - \Phi_s(sdf(\boldsymbol{p}_{i+1}))}{\Phi_s(sdf(\boldsymbol{p}_i))}, 0)$ is opacity, and $\boldsymbol{p}_i$ represents the sampled spatial point at $t_i$. $\Phi_s(x) = (1 + e^{-sx})^{-1}$ is a Sigmoid function, where $s$ is a learnable parameter which controls the smoothness of the transition at the surface.

Notably, the goal of novel view synthesis is to make an accurate prediction of the colors $\hat{C}$, and bend efforts to minimize the difference between the colors of ground truth images $C$ and the prediction $\hat{C}$:

$$C = \hat{C} = \sum_{i=1}^{n} w\left(t_i\right) \hat{c}\left(t_i\right). \tag{8}$$

In surface reconstruction tasks, what we concentrate more is the surface of the object rather than the color. In this way, the above formula can be rewritten as:

$$
\begin{aligned}
C &= \sum_{i=1}^{j-1} w\left(t_i\right) \hat{c}\left(t_i\right) + w\left(t_j\right) \hat{c}\left(\hat{t^*}\right) + w\left(t_j\right)\left(\hat{c}\left(t_j\right) - \hat{c}\left(\hat{t^*}\right)\right) + \sum_{i=j+1}^{n} w\left(t_i\right) \hat{c}\left(t_i\right) \\
&= w\left(t_j\right) \hat{c}\left(\hat{t^*}\right) + \varepsilon_{sample} + \sum_{\substack{i=1 \\ i \neq j}}^{n} w\left(t_i\right) \hat{c}\left(t_i\right) \\
&= w\left(t_j\right) \hat{c}\left(\hat{t^*}\right) + \varepsilon_{sample} + \varepsilon_{weight},
\end{aligned} \tag{9}
$$

where $\hat{sdf}(\hat{t^*}) = 0$, $t_j$ denotes the nearest sample point from $\hat{t^*}$, $\varepsilon_{sample}$ denotes the bias caused by sampling operation and $\varepsilon_{weight}$ denotes the bias caused by weighted sum operation of volume rendering. With Formula (2), it can be rewritten as:

$$w\left(t_j\right) \hat{c}\left(\hat{t^*}\right) + \varepsilon_{sample} + \varepsilon_{weight} = c\left(t^*\right), \tag{10}$$

$$\hat{c}\left(\hat{t^*}\right) = \frac{c\left(t^*\right) - \varepsilon_{sample} - \varepsilon_{weight}}{w\left(t_j\right)}. \tag{11}$$

There the total bias between the colors of object surface and estimated surface is:

$$\Delta c = \hat{c}\left(\hat{t^*}\right) - c\left(t^*\right) = \frac{(1 - w\left(t_j\right))c\left(t^*\right) - \varepsilon_{sample} - \varepsilon_{weight}}{w\left(t_j\right)}. \tag{12}$$

The relative bias is:

$$\delta c = \frac{\Delta c}{c\left(t^*\right)} = \frac{1}{w\left(t_j\right)} - 1 - \frac{\varepsilon_{sample} + \varepsilon_{weight}}{w\left(t_j\right) c\left(t^*\right)}. \tag{13}$$

When $w\left(t_j\right)$ approaches to 1, $\varepsilon_{weight}$ approaches to 0 and $\delta c$ approaches to $\varepsilon_{sample}/c(t^*)$. In this case, the total bias is only caused by discrete sampling, which is small (but still exists). Simulated weights of some existing neural reconstruction methods are shown in Fig. 3. As can be seen, it is nearly impossible to get right there in practice, especially without any geometric constraints. Furthermore, the problem becomes more intractable when dealing with cases of occlusion. Therefore, the weighted manner of volume rendering integral introduces a bias to implicit geometry modeling. Because the supervision of the whole network almost depends exclusively on the difference between rendered colors and ground truth colors, the

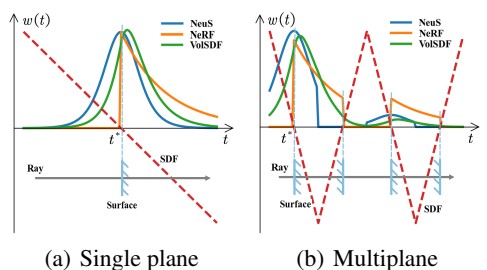

(a) Single plane       (b) Multiplane

Figure 3: Simulated weight in color rendering process of neural reconstruction methods.

bias would make it difficult to supervise the colors of surface and the SDF network, leading to a gap between the colors and the geometry.

A trivial solution is to directly supervise the geometry of the object. In this way, we design explicit supervision on the SDF network and geometry-consistent supervision with multi-view constraints.

## 3.2 Explicit supervision on SDF network

The SDF network, which estimates the signed distance from any spatial point to the surface of the object, is the key network that we need to optimize. So we propose an explicit supervision method on the SDF network to ensure its accuracy directly with points in 3D space.

For less extra cost, we use points generated by structure from motion (SFM) [29, 33] to supervise the SDF network. In fact, SFM is a canonical solution to compute the camera parameters of input images, where 2D feature matches $\boldsymbol{X}$ and sparse 3D points $\boldsymbol{P}$ are also generated as byproducts. Thus, these sparse 3D points can be used as "free" explicit geometry information. Approximately, we suppose that these sparse points are on the surface of the object. That is, the SDF values of the sparse points are zeroes: $sdf(\boldsymbol{p}_i) = 0$, where $\boldsymbol{p}_i \in \boldsymbol{P}$. In practice, after obtaining sparse 3D points, a radius filter is applied to exclude some outliers [46].

**Occlusion handling.** Because we focus on opaque objects, some parts of objects are invisible from view of a certain camera position. Therefore, there are only some of the sparse points visible for each view. For an image $I_i$ with camera position $\boldsymbol{o}_i$, the visible points $\boldsymbol{P}_i$ are consistent with feature points $\boldsymbol{X}_i$ of $I_i$:

$$\boldsymbol{X}_i = \boldsymbol{K}_i \left[\boldsymbol{R}_i | \boldsymbol{t}_i\right] \boldsymbol{P}_i, \tag{14}$$

where $\boldsymbol{K}_i$ is the internal calibration matrix, $\boldsymbol{R}_i$ is the rotation matrix and $\boldsymbol{t}_i$ is the translation vector for image $I_i$. The coordinates of $\boldsymbol{X}_i$ and $\boldsymbol{P}_i$ are all homogeneous coordinates. The scale index before $\boldsymbol{X}_i$ is omitted for simplicity. According to feature points of each image, we get visible points for each view and use them to supervise the SDF network while rendering image from the corresponding view.

**View-aware SDF loss.** While rendering image $I_i$ from view $V_i$, we use the SDF network to estimate SDF values for the visible points $\boldsymbol{P}_i$ of $V_i$ (see supplementary for the SDF loss by random sampling from sparse 3D points). Based on the approximation that the SDF values of sparse points are zeroes, we propose the view-aware SDF loss:

$$\mathcal{L}_{SDF} = \sum_{\boldsymbol{p}_j \in \boldsymbol{P}_i} \frac{1}{N_i} |\hat{sdf}(\boldsymbol{p}_j) - sdf(\boldsymbol{p}_j)| = \sum_{\boldsymbol{p}_j \in \boldsymbol{P}_i} \frac{1}{N_i} |\hat{sdf}(\boldsymbol{p}_j)|, \tag{15}$$

where $N_i$ is the number of points in $\boldsymbol{P}_i$ and $|\cdot|$ denotes the $L_1$ distance. It is worth noting that the loss we use to supervise the SDF network varies according to the view being rendered. In this way, the introduced SDF loss is consistent with the process of color rendering.

With the explicit supervision on the SDF network, our network could converge faster owing to the use of geometry prior. Besides, because the complex geometric structures with strong textures are the concentrated distribution areas of the sparse points, our method could capture more meticulous geometries.

## 3.3 Geometry-consistent supervision with multi-view constraints

With SDF loss, our network could capture complex geometric details with strong textures. Since the sparse 3D points mainly provide the explicit constraints on the areas with rich textures, large smooth regions still lack explicit geometry constraints. To go a step further, we design geometry-consistent supervision on the implicit surface with multi-view stereo constraints.

**Occlusion-aware implicit surface capture.** We use the implicit representation of the surface, and extract surface with the zero-level set of the implicit function. So the question is: Where is our implicit surface? According to Formula (3), the estimated surface is:

$$\partial \hat{\Omega} = \left\{ \boldsymbol{p} | \hat{sdf}(\boldsymbol{p}) = 0 \right\}. \tag{16}$$

We aim to optimize $\partial \hat{\Omega}$ with geometry-consistent constraints among different views. Because the number of points on the surface is infinite, we need to sample points from $\partial \hat{\Omega}$ in practice. To maintain consistency with the process of color rendering using view rays, we sample the surface points on these rays. As mentioned in 3.1, we sample $t$ discretely along the view ray and use the Riemann sum to obtain the rendered colors. Based on the sampled points, we use linear interpolation to get the surface points, which is similar to the root-finding used in [25, 26] to estimate the surface.

Specifically, with sampled point $t$ on the ray, the corresponding 3D point is $\boldsymbol{p} = \boldsymbol{o} + t\boldsymbol{v}$, and the predicted SDF value is $\hat{sdf}(\boldsymbol{p})$. For simplicity, we further represent $\hat{sdf}(\boldsymbol{p})$ as $\hat{sdf}(t)$, which is the function of $t$. We find the sample point $t_i$, the sign of whose SDF value is different from the next sample point $t_{i+1}$. The sample points set $T$ formed by $t_i$ is:

$$T = \left\{ t_i | \hat{sdf}(t_i) \cdot \hat{sdf}(t_{i+1}) < 0 \right\}. \tag{17}$$

In this situation, the line $t_i t_{i+1}$ intersects with the surface $\partial \hat{\Omega}$. The intersection points set $\hat{T}^*$ is:

$$\hat{T}^* = \left\{ t \mid t = \frac{\hat{sdf}(t_i)t_{i+1} - \hat{sdf}(t_{i+1})t_i}{\hat{sdf}(t_i) - \hat{sdf}(t_{i+1})}, t_i \in T \right\}. \tag{18}$$

The ray that interacts with the object may have more than one intersection with the surface. Specifically speaking, there may be at least two intersections. Similar to the SDF supervision mechanism, we just use the first intersection point along the ray considering the occlusion problem:

$$t^* = \operatorname{argmin} \left\{ t \mid t \in \hat{T}^* \right\}. \tag{19}$$

The selection of $t^*$ guarantees the sample points of the implicit surface are all visible for the corresponding view and makes the supervision consistent with the process of color rendering.

**Multi-view photometric consistency constraints.** We capture our estimated implicit surface, of which the geometric structures are supposed to be consistent among different views. Based on this intuition, we use the photometric consistency constraints in multi-view stereo (MVS) [9, 37, 10] to supervise our extracted implicit surface.

For a small area $s$ on the surface, the projection of $s$ on the image is a small pixel patch $q$. The patches corresponding to $s$ are supposed to be geometry-consistent among different views, except for occlusion occasions. Similar to patch warping in traditional MVS methods, we use the central point and its normal to represent $s$. For convenience, we represent the plane equation of $s$ in the camera coordinate of the reference image $I_r$:

$$\boldsymbol{n}^T \boldsymbol{p} + d = 0, \tag{20}$$

where $\boldsymbol{p}$ is the intersection point computed through Formula (19) and $\boldsymbol{n}^T$ is the normal computed with automatic differentiation of SDF network at $\boldsymbol{p}$. Then the image point $\boldsymbol{x}$ in the pixel patch $q_i$ of reference image $I_r$ is related to the corresponding point $\boldsymbol{x}'$ in the pixel patch $q_{is}$ of the source image $I_s$ via the plane-induced homography $\boldsymbol{H}$ [12]:

$$\boldsymbol{x} = \boldsymbol{H}\boldsymbol{x}', \boldsymbol{H} = \boldsymbol{K}_s (\boldsymbol{R}_s \boldsymbol{R}_r^T - \frac{\boldsymbol{R}_s (\boldsymbol{R}_s^T \boldsymbol{t}_s - \boldsymbol{R}_r^T \boldsymbol{t}_r) \boldsymbol{n}^T}{d}) \boldsymbol{K}_r^{-1}, \tag{21}$$

where $\boldsymbol{K}$ donates the intrinsic matrix, $\boldsymbol{R}$ donates the rotation matrix and $\boldsymbol{t}$ donates the translation vector. The index indicates which image the donation belongs to. To concentrate on the geometric information, we convert color images $\{I_i\}$ into gray images $\{I_i'\}$, and supervise our implicit surface with the photometric consistency among patches in $\{I_i'\}$ (see supplementary for RGB image settings).

**Photometric consistency loss.** To measure the photometric consistency, we use the normalization cross correlation (NCC) of patches in the reference gray image $\{I_r'\}$ and the source gray image $\{I_s'\}$:

$$NCC(I_r'(q_i), I_s'(q_{is})) = \frac{Cov(I_r'(q_i), I_s'(q_{is}))}{\sqrt{Var(I_r'(q_i))Var(I_s'(q_{is}))}}, \tag{22}$$

where $Cov$ denotes covariance and $Var$ donates variance. While rendering colors for an image, we use the patches which take the pixels being rendered as center and the patch size is $11 \times 11$. We take the rendered image as the reference image and compute NCC scores between its sampled patches and their corresponding patches on all source images. To handle occlusions, we find the best four of the computed NCC scores for each sampled patch following [10], and use them to compute the photometric consistency loss for the corresponding view:

$$\mathcal{L}_{photo} = \frac{\sum_{i=1}^{N} \sum_{s=1}^{4} 1 - NCC(I_r'(q_i), I_s'(q_{is}))}{4N}, \tag{23}$$

where $N$ is the number of sampled pixels on the rendered image. With the photometric consistency loss, the geometric consistency of the implicit surface among multiple views is guaranteed.

### 3.4 Loss function

During rendering colors from a specific view, our total loss is:

$$\mathcal{L} = \mathcal{L}_{color} + \alpha \mathcal{L}_{reg} + \beta \mathcal{L}_{SDF} + \gamma \mathcal{L}_{photo}. \tag{24}$$

$\mathcal{L}_{color}$ is the difference between the ground truth colors and the rendered colors:

$$\mathcal{L}_{color} = \frac{1}{N} \sum_{i=1}^{N} |C_i - \hat{C}_i|. \tag{25}$$

And $\mathcal{L}_{reg}$ is an eikonal term [11] to regularize the gradients of SDF network:

$$\mathcal{L}_{reg} = \frac{1}{N} \sum_{i=1}^{N} (|\nabla \hat{sdf}(\boldsymbol{p}_i)| - 1)^2. \tag{26}$$

In our experiments, we choose $\alpha$, $\beta$ and $\gamma$ as 0.1, 1.0 and 0.5 respectively.

## 4 Experiments

### 4.1 Experimental setting

**Datasets.** Following previous practices [43, 35, 42], we reconstruct surfaces from 15 scans of DTU dataset [1] to evaluate our method. DTU dataset has objects of various categories, which are quite different in terms of appearance and geometries. There are 49 or 64 images at a resolution of $1200 \times 1600$ in each scan with camera parameters. We also test on 7 challenging scenes from the low-res set of the BlendedMVS dataset [41] (CC-4 License). Scenes in BlendedMVS have various numbers of views and camera parameters. The scenes are captured by images at a resolution of $768 \times 576$, and the numbers of views vary from 31 to 143. We evaluate our reconstructed surfaces on DTU dataset with the Chamfer Distance provided by DTU evaluation metrics [1]. For the BlendedMVS dataset, we show the visual effects of the reconstructed surfaces.

**Baselines.** To better evaluate our method, we compare it with the-state-of-art learning-based methods and the traditional reconstruction method, colmap [30]. For learning-based methods, we compare with IDR [43], VolSDF [42], NeuS [35] and NeuralWarp [7]. For colmap, we use the reconstructed surface with trim parameter 7 (the best performance) [26].

**Implementation details.** Similar to [43, 35, 42], the SDF network is modeled by an 8-layer MLP with 256 hidden units and a skip connection in the middle. It is initialized by the geometric initialization presented in [2]. The radiance network is parameterized by a 4-layer MLP with 256 hidden units. Positional encoding [20] is applied to 3D location with 6 frequencies and to viewing direction with 4 frequencies. We sample 512 rays per batch and follow the hierarchical sampling strategy in NeuS [35] to sample points for each ray. We train our model for 300k iterations for around 16 hours on a single NVIDIA RTX2080Ti GPU. After network training, a mesh can be extracted from the SDF in a predefined bounding box by the Marching Cube [19] with the volume size of $512^3$.

### 4.2 Comparisons

We compare the reconstruction quality of our method and baselines on DTU dataset. Table 1 shows the quantitative results. Notably, our method outperforms baselines by a large margin. Specifically, it outperforms state-of-the-art neural implicit surfaces learning methods by over 25% and outperforms the traditional method colmap by 22%. As shown qualitatively in Fig. 4, our method achieves high-quality surface reconstruction in both complex thin structures and large smooth regions. For example, our method can recover abrupt depth changes in Scan 37 and reconstruct planar structures in Scan 24 and 40. To test the capability of handling

| Scan | with mask | | without mask | | | | |
| | IDR | NeuS | VolSDF | NeuS | NeuralWarp | colmap | Ours |
|---|---|---|---|---|---|---|---|
| 24 | 1.63 | 1.15 | 1.14 | 1.37 | 0.49 | 0.45 | **0.375** |
| 37 | 1.87 | 0.95 | 1.26 | 1.21 | 0.71 | 0.91 | **0.537** |
| 40 | 0.63 | 0.80 | 0.81 | 0.73 | 0.38 | 0.37 | **0.336** |
| 55 | 0.48 | 0.39 | 0.49 | 0.40 | 0.38 | 0.37 | **0.357** |
| 63 | 1.04 | 1.26 | 1.25 | 1.20 | **0.79** | 0.90 | 0.800 |
| 65 | 0.79 | 0.72 | 0.70 | 0.70 | 0.81 | 1.00 | **0.454** |
| 69 | 0.77 | 0.69 | 0.72 | 0.72 | 0.82 | 0.54 | **0.408** |
| 83 | 1.33 | **0.94** | 1.29 | 1.01 | 1.20 | 1.22 | 1.032 |
| 97 | 1.16 | 1.14 | 1.18 | 1.16 | 1.06 | 1.08 | **0.843** |
| 105 | 0.76 | 0.77 | 0.70 | 0.82 | 0.68 | 0.64 | **0.548** |
| 106 | 0.67 | 0.66 | 0.66 | 0.66 | 0.66 | 0.48 | **0.460** |
| 110 | 0.90 | 1.35 | 1.08 | 1.69 | 0.74 | 0.59 | **0.473** |
| 114 | 0.42 | 0.39 | 0.42 | 0.39 | 0.41 | 0.32 | **0.294** |
| 118 | 0.51 | 0.51 | 0.61 | 0.49 | 0.63 | 0.45 | **0.355** |
| 122 | 0.53 | 0.52 | 0.55 | 0.51 | 0.51 | 0.43 | **0.345** |
| mean | 0.90 | 0.82 | 0.86 | 0.87 | 0.68 | 0.65 | **0.508** |

Table 1: Results on DTU scenes. The surfaces produced by colmap are trimmed with trimming value 7.

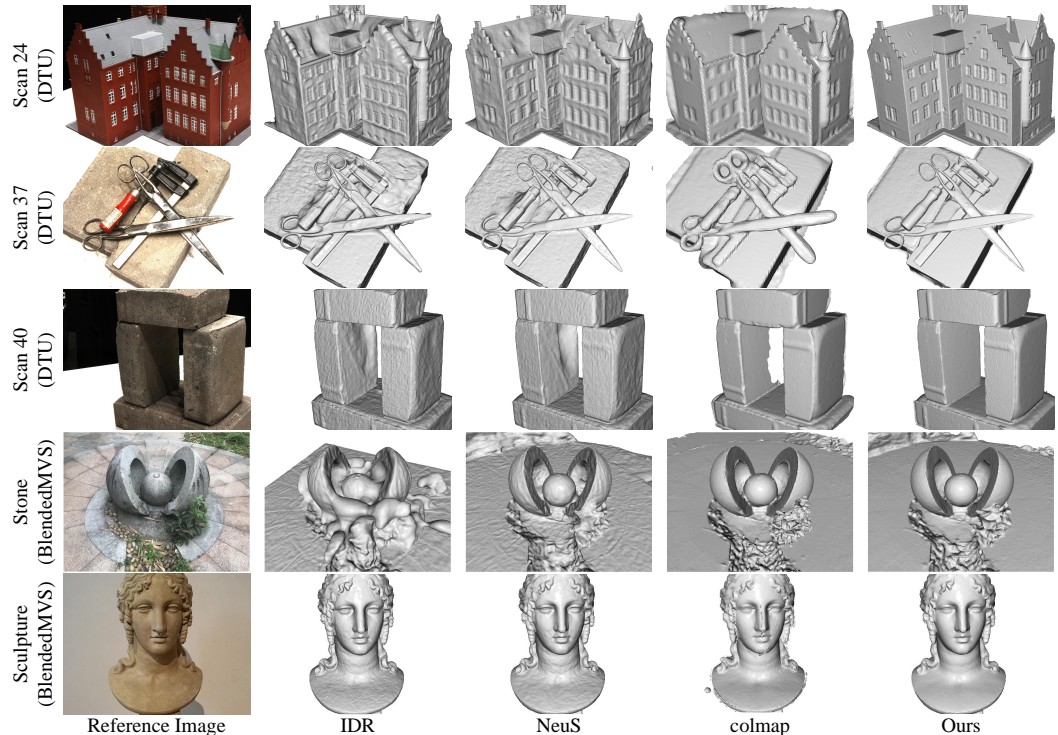

Figure 4: Surfaces reconstructed on DTU and BlendedMVS. We use NeuS trained with mask supervision and colmap with trimming value 7 (see supplementary for comparison with NeuralWarp).

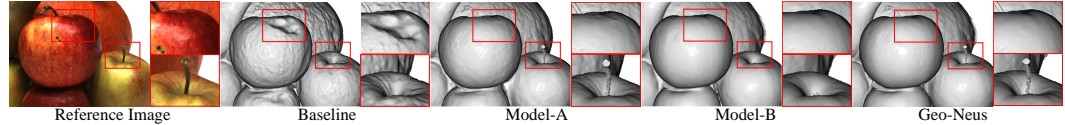

Figure 5: Surface quality of ablation models.

various scenes, we test on 7 challenging
scenes of the BlendedMVS dataset. Qualitative results in Fig. 4 show that our method yields more smooth and consistent surface quality than other methods.

### 4.3 Analysis

**Ablation study.** To evaluate the effect of our proposed contributions, we conduct an ablation study on DTU dataset. NeuS is adopted as our baseline. Different modules are progressively added to the baseline to investigate their efficacy. Results are reported in Table 2. We see that, with very sparse 3D supervision on SDF networks, Model-A has begun to outperform colmap (0.62 vs 0.65). This demonstrates that explicit SDF optimization is very beneficial to improve geometries. With the proposed photometric consistency loss, Model-B can optimize SDF networks more completely, leading to much more performance improvement. Fig. 5 shows how the proposed loss functions improve the surface quality. Model-A reconstructs the apple stem finely but the surface is not smooth enough. Model-B reconstructs the smooth surface but the apple stem is lost. That is, the SDF loss is better to improve the reconstruction of complex thin structures, while the photometric loss is better for the reconstruction of large smooth regions. Moreover, our full model, Geo-Neus absorbs their individual advantages and achieves the best performance.

**Geometry bias of volumetric integration.** To further investigate the geometric bias of volumetric integration, we render the depth images from a particular pose in a similar fashion to rendering RGB pixels [20] (see supplementary for details), and then use the depth images to construct sparse 3D points and photometric consistency constraints. NeuS is also used as the baseline. Comparison results are shown in Table 3. As can be seen, compared with baseline (0.87), multi-view geometry

| Method | $\mathcal{L}_{color}$ | $\mathcal{L}_{SDF}$ | $\mathcal{L}_{photo}$ | mean |
|---|---|---|---|---|
| Baseline | ✓ | | | 0.87 |
| Model-A | ✓ | ✓ | | 0.62 |
| Model-B | ✓ | | ✓ | 0.54 |
| Geo-Neus | ✓ | ✓ | ✓ | **0.51** |

Table 2: Ablation study on DTU scenes.

| Constraint | Setting | mean |
|---|---|---|
| Sparse 3D points | Depth integral | 0.85 |
| | SDF location | **0.62** |
| Photometric consistency | Depth integral | 1.08 |
| | SDF location | **0.57** |

Table 3: Comparison results between depth integral and SDF location.

constraints with depth integral bring little performance improvement or even degradation. It is a remarkable fact that photometric consistency supervision with depth integral surface location could not converge because of the initial immense bias while the SDF location model converges smoothly. As an alternative, we train these two models based on the baseline model pretrained with 200k iterations. The result of the depth integral model still degrades compared with the baseline. This verifies the existence of geometric bias in volumetric integration. With our proposed SDF-oriented optimization, surface reconstruction quality can be significantly boosted.

**Convergence speed.** We further study the convergence speed of our proposed method, Geo-Neus, and baseline, NeuS. As shown in Fig. 6, our method converges rapidly from scratch and becomes stable after 200k iterations. In contrast, NeuS cannot extract the reasonable surface from SDF networks in the beginning and starts to become stable after 250k iterations. This demonstrates that our proposed explicit SDF optimization also improves the efficiency of neural surfaces learning by volume rendering, reducing the training time from around 16 hours to around 10 hours.

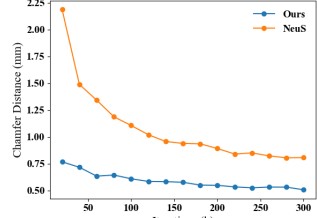

Figure 6: Convergence speed.

**Limitation.** We show failure cases of our method on scenes with strong specular highlights and transparent objects. Fig. 7(a) shows our reconstructed surfaces on the scene with strong specular highlights, DTU scan77. In this case, the strong specular highlights lead to large view-dependent effects, making the multi-view photometric consistency loss unable to reliably measure multi-view geometry constraints. Thus, our method cannot produce satisfactory surfaces in this case. In addition, Fig. 7(b) shows our reconstructed surfaces on the scene with transparent objects [13]. As proposed in Sec. 3.1, we assume the target objects are all opaque and solid. For transparent objects, the proposed bias between rendered colors and implicit geometry is invalid. Furthermore, the geometric loss we proposed may not work well, just like the photometric consistency used in traditional reconstruction methods.

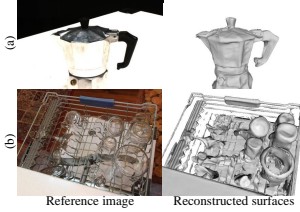

Reference image    Reconstructed surfaces

Figure 7: Failure cases.

## 5 Conclusion

We have proposed Geo-Neus, a new method to perform neural implicit surfaces learning by enforcing explicit SDF optimization. In our paper, we first provide the theoretical analysis that there exists a gap between volume rendering integration and neural SDF learning. With this theoretical support, we propose to explicitly optimize neural SDF learning by introducing two multi-view geometry constraints: sparse 3D points in structure from motion and photometric consistency in multi-view stereo. In this way, Geo-Neus produces high-quality surface reconstruction in both complex thin structures and large smooth regions. Therefore, it outperforms the state-of-the-arts by a large margin, including both traditional and neural implicit surfaces learning methods. We note that although our method greatly improves reconstruction quality, its efficiency is still limited. In the future, it will be interesting to explore accelerating neural implicit surfaces learning by volume rendering through super-fast per-scene radiance field optimization methods [34, 24]. We don't see an immediate negative societal impact of our work, but accurate 3D models may be used from malevolence.

**Acknowledgments** This work was in part supported by the National Natural Science Foundation of China under Grants 62176096 and 61991412, the Data Science and Artificial Intelligence Research Center (DSAIR), School of Computer Science and Engineering, Nanyang Technological University and the A*Star Center for Frontier AI Research (CFAR).

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
