# OpenReview forum: "Geo-Neus: Geometry-Consistent Neural Implicit Surfaces Learning for Multi-view Reconstruction"
_NeurIPS.cc/2022/Conference — NeurIPS 2022 Accept_

### Official Review · Reviewer_ee2G · 2022-06-28

**Rating:** 6
**Confidence:** 4
**Soundness:** 3 good
**Presentation:** 2 fair
**Contribution:** 3 good

**Summary:**

The authors of this manuscript propose Geo-NeuS, a novel neural implicit 3D reconstruction method.
They combine a SDF-based neural implicit surface representation, that is usually only optimized with the reconstruction loss, with explicit multi-view constraints.
More specifically, they incorporate supervision in the form of i) the sparse point cloud obtained from SfM, ii) photometric consistency from classic multi-view stereo.
They first theoretically analyze the difference of volume rendering and surface rendering-based approaches wrt. 3D reconstruction, and then show experimentally that adding the proposed constraints leads to better 3D reconstruction and more accurate surfaces.

**Questions:**

1.) Why is BlendedMVS not evaluated quantitatively?

2.) Why is the explicit SDF supervision (Sec. 3.2) implemented with occlusion handing and view-aware, instead of a simple loss via sampling in 3D space?

3.) Why are grey-scale images used for the photoconsistency loss? Could an ablation study help to support this design choice?

4.) Why did the authors use a patch size of 11x11?

5.) Where / For which type of surfaces do the proposed losses help in particular (e.g. specular surfaces, homogeneous areas, etc)?

**Limitations:**

The authors discussion on limitations and negative societal impact (L. 331 - 334) is quite limited. The authors could tackle more complex datasets / more complex scenes if the proposed system does not fail on DTU and the selected BlendedMVS scenes. What happens if photoconsistency is not given, e.g., because of strong specular highlights? How does the model perform in the presence of transparent surfaces? How could the model handle sparse set of inputs instead of the considered dense coverage of scenes? These could be starting points for an interesting limitation discussion.

**Strengths And Weaknesses:**

# Strengths

1.) The proposed method clearly improves quantitatively (Tab 1) as well as qualitatively (Fig 4) over the state-of-the-art.

2.) The proposed additional losses are well-grounded, not too complicated in general, and clearly have benefits for the task of 3D reconstruction (Tab. 2)

3.) The ablation study (Tab. 3) of applying the proposed losses either via expected depth maps or via surface points found via root finding is very interesting.

4.) While the findings of the theoretical analysis of volume vs surface rendering approaches for 3D reconstruction (Sec. 3.1) are expected, it is still valuable to have it.

5.) The manuscript is written clearly, has a good structure, and experimental evaluation is enough to show the benefits of the proposed system.

# Weaknesses

1.) BlendedMVS is not evaluated quantitatively, and I couldn't find an argument for this.

2.) It is not clear to me why the explicit SDF supervision (Sec. 3.2) is done with occlusion handling (L. 165) and view-aware (L. 173). It is only stated that "the introduced SDF loss is consistent with the process of color rendering" (L. 177 - 178). Instead, in every iteration, a subset of the sparse point cloud could be sampled and the loss can be applied on the 3D location without any occlusion reasoning etc. which seems simpler and more straight-forward. I believe a good reason / ablation study for this complicated setup is missing

3.) The described process of finding the surface intersection (L. 197ff) is very similar to proposed root-finding methods from ray marching-based approaches for neural implicit surfaces like [23] and a short note on this+citation on this would be helpful for the reader.

4.) The fact that the photometric consistency loss is only applied on grey-scale images (L. 211ff) is interesting, and an ablation study on this would be helpful.

5.) NCC is used as the phometric consistency metric. Have the authors investigated other measures as well? This could be an interesting ablation study (but not a must in this manuscript).

6.) It is not clear how the patch size of 11x11 (L. 227) was determined.

7.) The fact that Colmap's best trim parameter is 7 (L. 253) should be cited, e.g. [23].

8.) The visual ablation study (Fig 5) could be bigger with zoom-in windows to better see the differences, similar to Fig 1 of Supp Mat.

9.) Table 3 / "Geometry bias of volumetric integration": very interesting, but details are missing. Are here the expected depth maps used obtained via volume rendering? I think at least the supp mat should contain relevant formulas how the quantities are obtained.

10.) Appendix C: Why is NeRF's rendering quality worse than NeuS / Geo NeuS?

11.) Would be interesting to further discuss or which situations the losses help in particular, e.g. mostly for specular areas?

12.) Fig 4 caption typo: NueS -> NeuS

---

> ### Author Response · Authors · 2022-08-02
> **Response**
>
> We thank the reviewer for the detailed comments and constructive suggestions. Below are our responses to the questions.
>
> **R4-Q1. Quantitative evaluation on BlendedMVS.** Unlike DTU which provides accurate point clouds obtained by 3D scanner as groundtruth for evaluation, there are only meshes constructed by MVS pipelines from images in BlendedMVS. Following NeuS and UNISURF, we just provide the qualitative results on BlendedMVS.
>
> **R4-Q2. SDF loss by random sampling from sparse 3D points.** We use visible points of SFM points to supervise when rendering a specific view, the aim of which is to make the geometry supervision consistent with the process of color rendering. This consistency make the optimization of color loss and SDF loss aim on the same region of surface, which provides a mutual guarantee. For further exploration, we do the ablation study on scan24 and scan37 of DTU with random sampled points from the sparse points. The number of sampled points is set to be the average of points in our experiments. The reconstruction results with random sampling get 0.43 and 0.58, which are worse than those with view-aware sampling (0.38 and 0.54). This validates the effectiveness of view-aware sampling. We have added this experiment in our supplementary material Sec. C.5.
>
> **R4-Q3. Photometric consistency loss by RGB images.** We use the grey-scale images for less time and memory consumption. On a on NVIDIA 2080TI, the training time with grey-scale images is about 16h while that with RGB images is about 24h. The extra GPU memory consumption is 0.6G and 1.1G respectively. We do the ablation study on DTU scan24 to compare photometric consistency effect with grey-scale images and RGB images. It shows that the network degrades with RGB images (RGB: 0.44 vs. Grey-scale: 0.38). We suppose that gray images may reflect more geometric information. We have add this experiment in our supplementary materail Sec. C.6.
>
> **R4-Q4. Other measures for photometric consistency loss.** We try to use SSIM to compute the photometric loss. The result becomes 0.408 on scan 24, which is a little worse than that of using NCC, 0.375. In COLMAP, the authors use an improved version of NCC, bilateral weighted NCC. Due to the time limit, we will explore this in the future. Related discussions have been added in our supplementary material Sec. C.7.
>
> **R4-Q5. How to determine the patch size of $11\times11$?**  Traditional MVS methods, such as COLMAP [28], Gipuma [9], ACMM [34], generally use the patch size of $11\times11$ to compute patch similarity. Following these practices, we also use this patch size to compute NCC scores.
>
> **R4-Q6. Details of volumetric integration.** Thanks for your helpful reminder. The expected depth maps we use are obtained by the depth integration, which is similar with color integration. In volume rendering, the expected color is calculated as: $\hat{C}= \sum_{i=1}^n{w\left( t_i \right) \hat{c}\left( t_i \right)}$. Similarly, we calculate the expected depth $\hat{d}$ as:
> $\hat{d}= \sum_{i=1}^n{w\left( t_i \right) d\left( t_i \right)}$. We have add these details in our supplementary materials Sec. A.
>
> **R4-Q7. Why is NeRF's rendering quality worse than NeuS/Geo-Neus?** NeuS and Geo-Neus use another rendering network, NeRF++, to model background. This enhances their rendering capability. VolSDF does not use anoher network to model background, thus achieving similar PSNR to NeRF. This makes the rendering quality of NeuS and Geo-Neus better than that of NeRF and VolSDF.
>
> **R4-Q8. Which situations do the proposed losses help in particular?** In our experiments, we find homogeneous areas are  particularly improved, such as scan40 in DTU, stone in BlendedMVS and etc. Besides, areas with occlusions are also improved a lot, such scan 37 in DTU, dog in BlendedMVS and etc.
>
> **R4-Q9. Missing references and typos.** Thank you for pointing out these issues. We have added the missing references and fix the typos in the revision accordingly.
>
> **R4-Q10. Limitations.** Thank you for your constructive suggestions. For scene with strong specular highlights and transparent objects, our method degrades in these situations. Visualization results are shown in our supplementary materail Sec. F.2. In addition, we also explore the potential of our method with sparse input views. We find that our method can still achieve satisfactory results (including geometry and rendering) in this case while NeuS degrades a lot. This shows the superiority of our proposed method. More details have been added in our supplementary material Sec. F.1.

---

> > ### Comment · Reviewer_ee2G · 2022-08-03
> > **References in main paper, limitation discussion**
> >
> > Thank you very much for the rebuttal!
> >
> > I believe that my raised questions are addressed. I would, however, suggest to at least provide references to the main paper referencing to the added experiments for Q2 and Q3 in the supp mat. Further, as one additional content page is added for the final version, I believe adding the failure analysis of Q10 to the main paper would make the paper significantly stronger.
> >
> > Thanks!

---

> > > ### Author Response · Authors · 2022-08-03
> > > **Thanks for your feedback!**
> > >
> > > We are very grateful for your constructive suggestions and help in improving the paper!  We will add the failure analysis and the references to the corresponding experiments in the final revision of our main paper.

---

### Official Review · Reviewer_Xndx · 2022-07-08

**Rating:** 5
**Confidence:** 4
**Soundness:** 3 good
**Presentation:** 2 fair
**Contribution:** 3 good

**Summary:**

This paper presents neural implicit surfaces learning by enforcing explicit SDF constraints. It analyzed the gap between volume rendering integration and implicit SDF learning. Based on that, it proposes an explicit surface point SDF supervision where the surface points are estimated with SFM. in addition, it locates the zero-level set of SDF networks and a multiview photometric consistency loss is proposed to explicitly supervise the training of SDF networks. The results demonstrate high quality 3d scene geometry reconstruction, even on thin structures and large flat regions.

**Questions:**

- For the multiview photometric consistency loss, it is not clear to me how the reference view is picked w.r.t. the source view. Is it pure random or within a perturbation range? Eq. 19 determines the ray intersection with the implicit surface. However, it seems to me it does not guarantee the intersection point will be visible for all the reference views. If the point occluded in the reference view, will the multiview constraint still be valid?

- It is also not clear to me why this approach does not require foreground mask while still being able to reconstruct the 3d scene without background.

- Please clarify how the ray sample weights w are converted from SDF values.

- Both the SDF loss and the multiview consistency loss seem reasonable to help regularize the geometry learning. However, it is not clear to me how they are directly linked to the motivation given in sec 3.1. Will it be possible to evaluate the bias w/wo the proposed losses?


**Ethics Review Area:**

["I don’t know"]

**Limitations:**

- The multiview photometric consistency loss does not consider the view dependent effects. it remains questionable for reflective or specular objects.

- It will be interesting to show the effect of number of reference cameras on the rendering quality. On one hand, with less reference views, the SFM might produce noisy result and affect the SDF guidance loss. It is not clear so far how the quality of SFM points affect the 3d scene reconstruction. On the other hand, multiview consistency loss might help regularize the neural field distribution and compared to baseline models, it might still converge with sparser training cameras.

**Strengths And Weaknesses:**

- The paper is well organized and written.
- Comprehensive experiments demonstrate the geometry improvements brought by the proposed 2 losses.
- It provides theoretical analysis for the geometry bias, although I would like to see an evaluation of the bias after introducing the 2 losses (see details below)
- Such warping patch losses have been explored in recent papers, such as NeuralWarp and MVCGAN, CVPR 2022

---

> ### Author Response · Authors · 2022-08-02
> **Response**
>
> We thank the reviewer for the detailed comments and constructive suggestions. Below are our responses to the questions.
>
> **R3-Q1. Evaluation of the bias with 2 proposed losses.** In sec.3.1, we analyze the bias between color rendering and implicit geometry, which indicates that it is unreliable to depend solely on rendering loss to reconstruct accurate surfaces. That motivates us to introduce SFM points and photometric consistency to supervise the geometry explicitly.  We try to evaluate the bias with 2 proposed losses and details can be found in our supplementary material Sec. G. With the help of proposed losses, the network could better simulate the real color field. But the sample bias and the weight bias still exist because of volume rendering. Our experiments show that the rendering quality of Geo-Neus is not improved compared with Neus owing to the integration effect of volume rendering. The predicted colors of surface (where the predicted SDF is 0) w/ and w/o the proposed losses are shown in Fig.1(b) in our paper. The geometric losses we propose could relieve the bias between color rendering and the implicit surface. Theoretical quantitative analysis of the bias is a very meaningful future work.
>
> **R3-Q2. Comparison with NeuralWarp and MVCGAN.** The difference between our method and NeuralWarp is shown in R2-Q1. MVCGAN adopts depth integral to represent surface points and then enforces multi-view geometry constraints on these surface points. As we discussed in Sec. 3.1, this introduces bias for true geometry modeling. To address this problem, we directly locate the zero-level set of SDF networks to represent the surface points and enforce explicit geometry constraints on these surface points. The experiments in geometry bias of volumetric integration verify the existence of this bias and show that our design can address this problem and achieve much better results.
>
> **R3-Q3. The selection of the reference view and source views and the visibility of intersection points.** The currently rendered image is selected as the reference image. Based on the sparse 3D points, we follow COLMAP to compute the total number of sparse 3D points observed by each image pair and their corresponding triangulation angles. When an image pair has more than 75\% of these triangulation angles below $5^{\circ}$, we remove this image pair. At last, we select the top 9 source views in terms of the number of co-visible 3D points. Since the currently rendered image is selected as the reference image, Eq. 19 shows that we select the nearest intersection points to represent the surface points for the reference image. This guarantees the intersection points are visible for the reference image. However, these intersection points may be not visible for all source views. To handle occlusions, we follow [9] to find the best four NCC scores to compute the photometric consistency loss.
>
> **R3-Q4. How to reconstruct the 3D scene without background?** Following the previous practices of IDR, VolSDF, NeuS and NeuralWarp, we use the visual hull of objects (defined by the segmentation masks of IDR) to reconstruct the 3D scene without background. Also, for fair comparison with previous neural reconstruction methods, we only evaluate the reconstruction inside the visual hull.
>
> **R3-Q5. Clarification of the conversion from SDF values to ray sample weights $w$.** Following NeuS, $w_i$ is computed as $w_i=T_i\alpha_i$, where $T_i=\prod_{j=1}^{i-1}(1-\alpha_j)$. $\alpha_j=\text{max}(\frac{\Phi_s(sdf({\textbf {\textit {p}}}_i))-\Phi_s(sdf({\textbf {\textit p}}_i+1))}{\Phi_s(sdf({\textbf {\textit {p}}}_i))},0)$. $\phi_s(x)=(1+e^{-sx})^{-1}$ is a Sigmoid function, where $s$ is a learnable parameter which controls the smoothness of the transition at the surface. We will add these details in our revision.
>
> **R3-Q6. The performance under sparse input views.** Thanks for your constructive suggestions. We select 3 input views from scan 97, 106, 118 of DTU and train NeuS and our model from scratch. The Chamfer distances of our method on these three scans are 1.045, 0.782 and 0.855 respectively, which are better than those of NeuS, 1.74, 1.85 and 3.59. Visualization results are shown in our supplementary material Sec. F.1. This shows that our method can still reconstruct satisfactory surfaces in this case, demonstrating the effectiveness of our proposed method.

---

### Official Review · Reviewer_1oxk · 2022-07-11

**Rating:** 5
**Confidence:** 5
**Soundness:** 3 good
**Presentation:** 3 good
**Contribution:** 3 good

**Summary:**

This paper introduces a new method for geometry-consistent neural implicit surfaces learning. The proposed method includes a theoretical analysis of the gap between volume rendering and point-based SDF modeling and a solution by leveraging sparse points from SfM and utilizing patchmatch to provide geometry consistent supervision. This method is evaluated on DTU and BlendedMVS and achieves improvements compared with existing methods.

**Questions:**

1. What’s the key difference between NeuraWarp and Geo-NeuS? Both methods adopted patch-match based optimization for better geometry reconstruction. The authors should provide a detailed discussion between these two methods and show their strengths, especially in the introduction and related works.

2. Sparse points from SfM are utilized to supervise the SDF network, where the sparse points with a radius filter are supposed on the surface (line 162). However, in practice, the sparse points from SfM are usually noisy, and wrongly incorporating these points with large errors may decrease the reconstruction quality.

3. The authors claim that discrete sampling can cause bias (line 138) and linear interpolation is adopted to get surface points (line 195). What’s the error of the extracted surface points by these two methods, i.e., the proposed method of Geo-NeuS and the hierarchical sampling method in NeuS? This experiment may support the author’s assumption and theoretical analysis.

4. The authors are encouraged to conduct a qualitative comparison between NeuralWarp and Geo-NeuS.

5. The training time is 16h with 300k iterations on NVIDIA 2080TI (line 259) while the training time of NeuS reported in its original paper on NVIDIA 2080TI is also about 16h. However, Geo-NeuS requires extra computation for depth and patch match supervision. I’m not sure whether this number is correct. Besides, what about the accuracy with fewer iterations, as discussed in line 313, such as 150k or 200k iterations?



**Limitations:**

Limitations or failure cases are not discussed in this paper.

**Strengths And Weaknesses:**

In this paper, the main contribution is incorporating depth supervision (from sparse points) and patch match supervision into NeuS’s optimization framework, which leads to better reconstruction results. It seems that the improvement mainly comes from integrating patch match, shown in Tab. 2, which has been demonstrated by NeuralWarp. So the discussion about these two methods is expected.

The paper is well organized, and the evaluations are sufficient to support the proposed method.

---

> ### Author Response · Authors · 2022-08-02
> **Response**
>
> We thank the reviewer for the detailed comments and constructive suggestions. Below are our responses to the questions.
>
> **R2-Q1. Comparison with NeuralWarp.** Thanks for the helpful reminder of discussion between Geo-Neus and NeuralWarp. We will add the discussion in our revised paper.
> NeuralWarp is an exploration of the use of patch-match on neural surface reconstruction. It combines volumetric rendering with a patch warping integration  technique, which aggregates colors from points sampled along the camera ray from source views with patch warping. This way of patch aggregation is similar with volume rendering and shares the same sampled points and the same weights with those used by color integration. Note that NeuralWarp uses patch match with the color aggregation to optimize weights of samples points, and thus to optimize the geometry indirectly. As we analyze in Sec.3.1, this kind of color integration operation will cause bias in the colors and the geometry. Therefore, NeuralWarp could not be trained from scratch and relies on the pre-trained model of VolSDF.
> Considering the bias analyzed in sec.3.1, we propose multi-view photometric consistency supervision to directly optimize the geometry represented by the SDF network. We locate the predicted surface of the SDF network using SDF-based interpolation and use patch match to measure the photometric consistency among neighboring views. In this way, Geo-Neus can be trained from scratch and get better performance.  We will add these related discussions in our revision.
>
> **R2-Q2. Noisy SFM points.** In our work, we use a radius filter with strict parameters to remove sparse points with large errors. This reduces the influence of sparse points with large errors as much as possible.  Besides, existing SFM techniques could reconstruct sparse points with pixel-level reprojection errors, which to some extent guarantees the accuracy of sparse points. Moreover, our method also leverages the color rendering loss and multi-view photometric consistency loss to implicitly/explicitly supervise the SDF network. This also guarantees our reconstruction quality. For further exploration, we use raw points obtained by SFM to do the ablation study on 3 scans of DTU (scan 24, 37 and 40), and get the accuracy 0.42, 0.63 and 0.34 respectively. It can be seen, our methods could also perform well with noisy sparse points directly from SFM. This shows the robustness of our method. Details of the ablation study can be found in our revised supplementary material Sec. C.3.
>
> **R2-Q3. The error of extracted surface points by linear interpolation and hierarchical sampling.** Thanks for your constructive suggestions. We replace the linear interpolation by the hierarchical sampling to extract surface points and retrain our model on DTU scan24 and scan37. The quantitative results by the hierarchical sampling are 0.537 and 0.677, which is worse than those of the linear interpolation (0.375 and 0.537). This further validates the gap between the volume rendering and SDF modeling, supporting our assumption and theoretical analysis. We have added this experiment in our supplementary material Sec. C.4.
>
> **R2-Q4. Qualitative comparison with NeuralWarp.** Thanks for your suggestion about qualitative comparison with NeuralWarp. We have added the qualitative comparison with NeuralWarp in our supplementary material Sec. E.2.
>
> **R2-Q5. Time consumption and accuracy with fewer iterations.** We find the time consumption problem may be caused by different cpus we use. We reevaluated the time consumption on scan 24 on our own device and found the time consumption of NeuS is about 14h 48min while Geo-Neus gets about 16h 14min. For accuracy, Geo-Neus could get 0.57 with 150k iterations on average and 0.54 with 200k. NeuS gets 0.94 and 0.89 respectively.
>
> **R2-Q6. Limitation and Failure cases.**  Our assumption in sec.3.1 is that the target objects are all opaque and solid. For transparent objects, the bias we talk about is invalid and the proposed supervision may not work well, just like the photometric consistency used in traditional reconstruction methods. For further exploration, we do experiments on glass bottles of Dex-Nerf dataset[C1] to test the performance of our method on transparent objects. The results validate our above analysis. We have added the related discussions in our supplementary Sec. F.2.
> [C1] Dex-NeRF: Using a Neural Radiance field to Grasp Transparent Objects, Jeffrey Ichnowski*, Yahav Avigal*, Justin Kerr, and Ken Goldberg, Conference on Robot Learning (CoRL), 2020

---

### Official Review · Reviewer_b5Ve · 2022-07-17

**Rating:** 5
**Confidence:** 3
**Soundness:** 3 good
**Presentation:** 2 fair
**Contribution:** 2 fair

**Summary:**

Implicit representation has become a popular technique for 3D scene reconstruction. Existing methods do not utilize the multi-view geometry constraints in the learning. In this paper, by leveraging sparse geometry from structure-from-motion (SFM) and photometric constraints in multi-view stereo.

The paper compares the proposed method with both classical (colmap [28]) and recent deep learning baselines (IDR [40], VolSDF [39], NeuS [33], and NeuralWarp [7]).


**Questions:**

The proposed method uses an 8-layer MLP for generating features from 3d points. We can also use a permutation-invariant method such as pointNet to generate features from 3D points. All the baselines are somewhat constrained to IDF-style algorithm. It would be good to also consider other baselines like Point2Surf or DVR in addition to the baselines shown:


**Limitations:**

The paper clearly mentions the limitations, especially the challenge in improving computationally efficiency.

**Strengths And Weaknesses:**

Strengths

1) The paper builds on top of other methods such as IDF [40] in handling multi-view geometry constraints while learning the model.

Weaknesses

1) Some of the top-performing SDF methods from poinclouds are not cited. Using the pointclouds from SfM methods, we can generate implicit 3D representations using methods such as points2surf:

Erler et al. Points2Surf: Learning Implicit Surfaces from Point Cloud Patches, ECCV 2020.

2) The chamfer distance shown in Table 1 has minor issues. For example, the proposed method is not the best performing method on scan 83. NeuS achieves better results.

3) The paper heavily relies on existing structure-from-motion pipelines  [51, 47, 22, 19] to compute the camera parameters.  In some sense, the MLP networks are essentially utilized to further refine the multi-view geometry constraints. Furthermore, if the classical methods fail on a scene it would be hard to use this method. The improvement in the results need to be explained more carefully since the geometry constraints are already implicitly incorporated while using the  structure-from-motion pipelines.

---

> ### Author Response · Authors · 2022-08-02
> **Response**
>
> We thank the reviewer for the detailed comments and constructive suggestions. We first discuss current neural implicit surface reconstruction methods and then show our responses to the questions.
>
> **R1. Discussion on current implicit surface reconstruction methods.**
> Current implicit surface reconstruction methods can be categorized into three types.
>
> 1.Reconstruction from scanned point clouds. Generally, these point clouds are uniformly scanned, thus representing relatively complete geometries. Some surface reconstruction methods, such as Points2Surf, ONet etc, can be used to reconstruct surfaces from those uniform and complete point clouds.
>
> 2.Reconstruction by multi-view stereo(MVS) methods. These methods take captured images as input. They first use SFM to recover camera parameters and obtain sparse 3D points. Then, MVS methods are used to compute dense point clouds. At last, the aforementioned surface reconstruction methods are applied on these dense point clouds to reconstruct surfaces.
>
> In the aforementioned methods, the uniform and complete point clouds need to be explicitly obtained for the subsequent surface reconstruction.  It should be noted that, these methods are rarely applied on the sparse point cloud produced by SFM to reconstruct surfaces.
>
> 3.Reconstruction from posed images by differentiable rendering. These methods are all based on NeRF framework. The NeRF-based methods, such as NeuS and Neuralwarp, also need to use SFM to compute camera parameters. However, they do not explicitly generate dense point clouds as MVS methods do. Based on the framework of differentiable rendering, they directly reconstruct surfaces from posed images with implicit representations, thus circumventing noises from point clouds.
>
> As a NeRF-based method, Geo-Neus inherits advantages of differentiable rendering frameworks. Besides, we use sparse points from SFM as an explicit geometry supervision. Although sparse points cannot represent complete geometry structures of scenes, they provide useful geometry information of rich-textured areas. Thus, with proposed SDF supervision, our method can outperform the SOTAs by a large margin.
>
> In general, Points2Surf uses either scanned point clouds like 1 or dense point clouds generated by MVS methods in 2 to reconstruct satisfactory surfaces. Indeed, Points2Surf can be also applied to the sparse points from SFM. However, these sparse points distribute very irregularly, which poses great challenges for existing reconstruction methods. Thus, the performance of Points2Surf degrades when handling sparse point clouds from SFM. The corresponding results are shown in our supplementary material Sec. B.
>
> **R1-Q1. Missing references of some SDF methods.**
> Thanks for your kind suggestions. We will add the missing references in our revision. As mentioned in R1, we also use Points2Surf to reconstruct surfaces from point clouds from SFM. The results show that the performance of Points2Surf degrades in this special case.
>
> **R1-Q2. Minor issues in Table 1.**
> Thanks for pointing out these issues. We have fixed them in the revision accordingly.
>
> **R1-Q3. Reliance on SFM pipelines and the role of MLP.**
> Thanks for your detailed comments. SFM is a key step in image-based 3d reconstruction methods. It is a routine to use SFM to compute camera parameters as inputs to the later process by MVS methods. Recently popular NeRF-based methods also use SFM to compute camera parameters. If the camera parameters obtained by SFM are inaccurate, it is hard for image-based methods to reconstruct promising surfaces. In fact, the SFM pipeline performs well in most reconstruction cases and thus makes the image-based methods widely used in practical 3d reconstruction tasks. Based on the frameworks of differentiable rendering, NeRF-based methods use MLPs to fit the color field and geometry field in 3d space. Built on this framework, we introduce a SDF loss with SFM points to help the SDF network to fit the geometry of objects better.
>
> **R1-Q4. Generating features from 3D points.**
> Thanks for your constructive suggestion about generating features from 3D points. Existing neural reconstruction methods are all based on NeRF and use the MLPs to fit the color field and the geometry field(such as SDF field). In our method, we use the sparse points in a geometric loss to directly supervise the SDF network instead of extracting features from them. We also tried to extract features from the sparse points by PointNet and made the SDF network conditioned on these features but got little improvement. Maybe the features are not representative because of the irregular neighbor regions caused by the extremely non-uniform distribution of sparse points. We are also very interested in exploring more geometric features from 3d points to help with the neural reconstruction pipeline for better reconstruction. We have added other baseline methods, such as Points2Surf and DVR, in our supplementary material Sec. B.

---

> > ### Comment · Reviewer_b5Ve · 2022-08-09
> > **Response**
> >
> > I would like to thank the authors for carefully addressing my concerns, especially w.r.t point2surf and feature generation from pointclouds. I have updated my recommendation. Thanks!

---

> > > ### Author Response · Authors · 2022-08-09
> > > **Thanks for your feedback!**
> > >
> > > Thank you very much for the careful review and constructive discussions. We will revise our paper following the comments made by all the reviewers.

---

### Meta-Review · Area_Chair_5n3g · 2022-08-25

**Recommendation:** Accept
**Confidence:** Certain

**Metareview:**

This paper introduces new and useful losses, presents a good experimental setup, supply analysis on the bias, and is clearly written. I encourage the authors to discuss similarities and differences to NeuraWarp and pointcloud->SDF methods in their revision.

**Award:**

No

---

### Decision · Program_Chairs · 2022-09-14

Accept